# Motivation matters: How enrollment motives shape doctoral experiences and career aspirations

**Nikita Smirnov****\*, Elena Tarasova**

Institute of Education, HSE University, Moscow, Russia

\* nsmirnov@hse.ru

## Abstract

This study examines the motivational profiles of Russian doctoral students and their implications for academic success, satisfaction, and career aspirations. We employ data from a nationwide survey of doctoral students at Russian universities (N = 1,267). Using latent class analysis we identify four distinct motivational profiles: Academic Orientation, Unconscious Motives, Topic Devoted, and Everything Everywhere All at Once. Our findings reveal that students with academic orientations exhibit higher satisfaction, greater confidence in successfully defending their dissertations, and a stronger intention to pursue research careers. In contrast, students with unconscious motives demonstrate lower satisfaction, reduced confidence, and a weaker inclination toward research career paths. These findings highlight the role of unclear enrollment goals or non-academic motivations, such as military service deferment, in shaping doctoral outcomes. Additionally, our analysis shows that engagement in research is the only aspect of the doctoral experience that fosters research career aspirations among students with initially non-academic motivations. These findings emphasize the need for Russian universities to reconsider admission policies, doctoral program structures, and academic support mechanisms to better accommodate the increasingly diverse expectations of doctoral students. The study contributes to the broader discourse on doctoral education by providing empirical insights into the relationship between motivation, institutional factors, and career trajectories.

## Introduction

While many students who enter doctoral programs are intrinsically motivated – driven by curiosity, intellectual challenge, or aspirations for an academic career – their motivations are far from uniform. Given the evolving role of doctoral education in society [1] and the increasing diversification of the doctoral student body [2], a more detailed examination of doctoral students' motives seems essential for a deeper

**Data availability statement:** Yes - all data are fully available without restriction; All relevant data are within the paper and its Supporting Information files.

**Funding:** NS and ET have both received support from the project "International academic cooperation" at HSE University. The funder had no role in study design, data collection and analysis, decision to publish, or preparation of the manuscript.

**Competing interests:** The authors declare that there exists no competing financial interest or personal relationships that could have appeared to influence the work reported in this paper.

understanding of the factors which could enhance the quality of doctoral education. It is especially important since motivation is one of the key factors influencing success and satisfaction in higher education in general, and in doctoral studies in particular [3,4]. As underrepresented groups increasingly enter doctoral programs [5–7], and as these programs are designed not solely for academic career trajectories [8–10], a holistic investigation of doctoral students' motivational profiles becomes even more critical. Moreover, if doctoral education is viewed as a key mechanism for reproducing human capital for academia and research-intensive sectors of the economy [11,12], studying motivations becomes particularly important.

Why do students enter doctoral programs? What aspects of the doctoral experience shape students' orientations toward research careers? Can doctoral students who initially enroll with non-academic motivation encounter experiences during their studies that foster a desire to pursue research in the future? We tend to discuss these issues using survey data from Russia.

Two key gaps exist in the current research on doctoral students' motivations. First, the geographical coverage is uneven, with Russia — ranked seventh among OECD (Organisation for Economic Co-operation and Development) countries in terms of doctoral graduates [13] — remaining underexplored. Second, existing studies tend to be narrowly focused and rarely examine how motivations relate to educational outcomes. Much of the literature addresses specific aspects, such as motivations for higher-degree research [14], professional doctoral education [15], psychological motivational profiles rather than overarching goals for pursuing a doctorate [16], or motivations within particular disciplinary fields [17]. Moreover, these studies are predominantly cross-sectional, documenting the prevalence of different motivation types without analyzing their impact on educational outcomes or the broader doctoral experience [17,18].

At the same time, the issue of doctoral student motivation is becoming increasingly pressing in the Russian context. First, despite the emerging institutional framework for industrial doctorates, doctoral education in Russia is currently structured solely as an academic program [19,20]. Meanwhile, the vast majority of doctoral students in Russia combine their studies with paid jobs [21], and many do not continue to work in the academic sector after graduation [22]. Understanding the motivations of those entering doctoral programs and exploring the factors that influence their career aspirations is therefore crucial.

Second, Russian doctoral education has faced a significant crisis over the past decade [23]. Declining demand for doctoral programs, high dropout rates, and a low proportion of defended dissertations are key indicators of this crisis [23–25]. As a result, the country risks a shortage of professionals for both academia and research-intensive sectors of the economy [26,27]. Understanding the motivations of Russian doctoral students could provide a foundation for addressing the challenges facing Russian doctoral education and mitigating this crisis.

This study aims to address three research questions

*RQ1:* What are the motivational profiles of Russian doctoral students?

*RQ2:* How are these motivational profiles related to the individual characteristics of doctoral students?

*RQ3:* Which elements of the doctoral experience foster the development of a research career orientation among students who initially enrolled with non-academic motives?

This study is driven by the conceptualization of doctoral education as a liminal space, a perspective widely discussed in the literature [28–31]. Originally emerging in anthropology and ethnography [32,33], the concept of liminality has been extensively applied to doctoral education research [30,34,35]. Liminality is characterized by several key features: the opacity of the liminal space, the unpredictability of the experience, a high degree of uncertainty, low levels of regulation, and the significant role of guides and mentors [32] — all of which are relevant to doctoral education.

In particular, this study focuses on the process of shedding pre-liminal properties: experiencing a liminal state involves relinquishing prior identities, altering one's perceptions of the social structure in which one is embedded, and adopting new "rules of the game" [32]. In the context of doctoral studies, this transformation involves reshaping students' academic habitus and fostering a stable research orientation [34,36]. This is particularly relevant in Russia, where non-academic motivation for pursuing a doctorate negatively impact the likelihood of successfully defending a dissertation [37]. Therefore, special attention is given to the environmental factors that facilitate the re-habituation of doctoral students with non-academic motivation and support them in completing their dissertations.

The novelty of this study lies, first, in its holistic examination of doctoral students' motivational profiles and, second, in its analysis of the relationship between motivation and various elements of the doctoral experience, as well as the exploration of factors that shape research-oriented career trajectories. Using data from a nationwide survey of Russian doctoral students, we perform latent class analysis to distinguish motivational profiles and further investigate the interplay of these profiles with various aspects of doctoral experience.

## National context

Certain characteristics of doctoral education in Russia may be linked to the motivational profiles of incoming students. First, Russian doctoral programs have traditionally maintained a strong academic orientation [23]. Until 2025, there was no differentiation between types of doctoral programs, nor were there frameworks for industrial or professional doctorates [38]. Moreover, the requirements for obtaining the PhD degree have been progressively tightened by the federal government [39]. At the same time, the evaluation of applicants' research qualification at the admission stage tends to be rather formal — even the most selective universities rarely employ holistic admission mechanisms [40]. As a result, Russia exhibits a high attrition rate and a low proportion of successfully defended dissertations. In 2023, 121,600 students were enrolled in doctoral programs in Russia, while only 14,146 graduated [41]. Among doctoral graduates, only 13% successfully defended their theses [42], a rate that has remained around 10% over the past decade.

Second, admission to doctoral programs in Russia is associated with significant social benefits. For instance, male students are eligible for military service deferment (as military service is compulsory for men in Russia) [43]. Additionally, doctoral students have the right to dormitory accommodation [44], which is significantly more affordable than typical housing costs. Students on state-funded placements receive a government stipend, although the amount remains low, averaging $35 per month for most specialties and $80 per month in certain fields.

Doctoral programs in Russia are implemented by three types of organizations: universities, research institutes, and organizations of continuing professional education. The vast majority of doctoral students are enrolled in universities (87% in 2022) [45], while the educational experiences of students in other types of institutions differ significantly [46]. This study focuses exclusively on doctoral students in the university sector.

## Conceptual framework: self-determination theory

This research employs Deci and Ryan's Self-Determination Theory (SDT) to analyze motivation, defined as the psychological forces driving behavior. SDT identifies three fundamental needs: autonomy, competence, and relatedness, which are fulfilled through supportive social environments [47,48]. Autonomy involves voluntary choice and self-control,

competence reflects self-efficacy and enthusiasm for challenges, and relatedness refers to feeling connected and accepted.

Motivation exists on a continuum from intrinsic to extrinsic, with intrinsic motivation driven by interest and enjoyment, while extrinsic motivation stems from external rewards or pressures. SDT further differentiates extrinsic motivation into four forms of regulation: integrated, identified, introjected, and external, ranging from the most autonomous to the least. Integrated regulation aligns external motives with personal values, as when doctoral students pursue a PhD for personal growth or long-term aspirations [14]. Identified regulation involves engaging in activities that support personal goals, such as acquiring professional skills [49]. Introjected regulation arises from internal pressures like social expectations, while external regulation is driven by rewards or obligations, such as seeking stable employment [14].

Amotivation, by contrast, reflects a lack of intention or awareness of goals and arises from feelings of incompetence or devaluing an activity [48]. Ryan and Deci emphasized that external motivations can become self-determined through processes of internalization and integration, making extrinsically motivated behaviors more autonomous [47]. Aligned with that, we are additionally exploring factors that might contribute to changes in doctoral students' career aspirations.

Doctoral students' motivations often combine intrinsic and extrinsic dimensions, forming unique motivational profiles that influence their commitment to research. Intrinsic motivation sustains students through genuine interest, while extrinsic motivations, such as career advancement or academic identity, complement these intrinsic drivers [50]. Throughout this study, we use "academic" and "intrinsic" interchangeably, as well as "non-academic" and "extrinsic". This study integrates SDT with the concept of motivational profiles to develop a typology of motives for enrolling in doctoral programs.

## Literature review

Previous studies suggest that motivation to pursue and succeed in doctoral education is a key factor influencing both doctoral program completion and successful dissertation defense [50–55]. However, doctoral students' motivation for pursuing a doctorate vary across disciplines. Students in the humanities and social sciences are more often driven by intrinsic motivations (interest in doctoral education itself) and aspirations for academic careers, whereas those in science and technology fields may prioritize labor market goals [18,56].

Researchers usually distinguish intrinsic and extrinsic motivation using Self-Determination Theory [47,48]. Intrinsic motivation, such as interest in learning and research, is strongly associated with higher academic performance and leads to higher educational outcomes [50,53]. Shin et al. [57] found that students aiming for an academic career – who are typically driven by intrinsic motivation – are more satisfied with their learning process and handle the stress of academic workloads more effectively, recognizing the impact of their dissertation on their professional advancement.

However, extrinsic or non-academic motives also play a significant role. These motives include career advancement outside academia, social benefits, financial incentives, and even pragmatic considerations such as avoiding military service or securing accommodation [58,59]. According to a recent study on Russian doctoral students, a total of 38% of respondents believed that pursuing doctoral studies would support career development outside academia, 33% enrolled to continue professional education, 23% wished to remain in the university environment, 8% cited access to university housing, and 25% of male respondents noted military service deferment as a reason to enroll [59]. Despite the prevalence of mixed motives, research suggests that non-academic extrinsic motivation is sometimes perceived by the academic community as misaligned with the primary goals of doctoral education [55].

Motivation to pursue doctoral studies may at the same time include intrinsic interests in knowledge and research, and extrinsic career aspirations, external influences such as family or mentors [2,14,17,60]. According to Bekova et al. [59], nearly half of Russian doctoral students (55%) reported mixed motives for pursuing a PhD, while 31% had purely academic motives and 14% had non-academic motives. Moreover, based on interviews with doctoral graduates, it has been demonstrated that the presence of both personal motivation (such as a desire to engage in research, an interest in learning, or a dream of earning a doctoral degree) and professional motivation (such as aspirations for career advancement

or financial rewards) is important for fostering persistence and success among doctoral candidates [54]. These motivations are rarely static; they can evolve over time due to contextual and environmental factors. For example, supportive supervision and collaborative research environments can foster intrinsic motivation [61]. However, although previous studies highlight the overlap of students' motives, their conceptualization often fails to account for the complex nature of motivation. For instance, Zhuchkova & Bekova [37] and Bekova [62] include motivations of Russian doctoral students as control variables but use binary coding (only academic vs. non-academic motives). In contrast, we try to adopt a holistic approach, combining single reasons to enroll into complex motivational profiles.

In conclusion, while academic motivation has consistently been linked to higher success rates in doctoral studies, non-academic motivation is becoming increasingly prevalent [58]. Bekova et al. [59] highlight a significant shift in doctoral students' career aspirations, reflecting the diversification of professional trajectories beyond academia – including careers in industry, government, and other sectors [49]. However, the growing emphasis on extrinsic goals suggests that the traditional academic trajectory is no longer the sole or even the primary aspiration for many doctoral students, highlighting the need to adapt doctoral training to these evolving professional landscapes.

## Data analysis

### Sample and data collection

This study is based on data from a nationwide survey conducted as part of the "Monitoring of Education Markets and Organizations" (MEMO) project on behalf of the Russian Ministry of Science and Higher Education [63]. The survey targeted doctoral students at Russian universities during the 2021–2022 academic year and was administered online between May 31 and August 31, 2022, through an online self-administered survey (Computer-Assisted Web Interviewing, CAWI). The sample was constructed using a quota-based approach, stratified by region and type of organization. The survey invitation was disseminated via the Integrated Analytical System "Monitoring" to universities and research institutions in Russia, accompanied by an official endorsement letter from the Ministry of Science and Higher Education. To increase response rates, reminder calls were made to institutional representatives during the data collection period. Within each institution, representatives were instructed to distribute the survey link in accordance with predefined quotas. Compliance with the quotas was monitored in real time via built-in counters in the survey platform. In cases of underrepresentation of specific quota categories, follow-up calls were made to facilitate additional responses.

Participation was voluntary and anonymous, with informed consent obtained by checking the box at the first screen of the survey. The study received approval from the Institutional Review Board at the Higher School of Economics (HSE University). The authors accessed the data on November 28, 2024.

A total of 2,392 doctoral students completed the questionnaire. For this analysis, participants from research institutes, as well as those specializing in medicine and agriculture, fifth-year students, and students in mixed or employer-funded programs, were excluded due to their small representation and distinct characteristics, which could bias the results.

After data cleaning, the final sample consisted of 1,267 doctoral students from 154 universities. Among them, 54% were female, 84% were enrolled full-time, and 75% studied tuition-free. The most common field of study was humanities (30% of the sample), followed by social sciences (26%), mathematics and Earth sciences (23%), and engineering and technical sciences (22%). Regarding the year of study, most participants were first-year students (37%), while the smallest group consisted of fourth-year students (10%). Additionally, 43% of the sample studied at leading universities, including federal universities, national research universities, and institutions participating in excellence programs.

### Motivational profiles of doctoral students

To describe enrollment motives, the participants answered the following multiple-choice question: "*What were your goals for enrolling into the doctorate?*". The logic behind proposed options is described in [59]. The results are presented at Fig 1.

**Motive**

| Motive | | |
|---|---|---|
| To receive PhD degree | | 87% |
| To develop research skills | | 64% |
| To develop teaching skills | | 48% |
| To continue research in the field of interest | | 44% |
| To move forward in academic career | | 40% |
| To get the deferment from the army (male only) | | 29% |
| To employ in the university / scientific organization | | 28% |
| To move forward in non-academic career | | 18% |
| To get an opportunity to travel abroad | | 12% |
| To receive the diploma without defending the thesis* | | 9% |
| To get a place in a dormitory | | 4% |

*Prior to the 2021-2022 reform in Russia, doctoral students could compete their program without defending a thesis (if they completed all courses and passed the exams). Now graduation requires having a prepared thesis text and sucessfully passing the pre-defence procedure.

**Fig 1. Prevalence of different admission motives.** Question: "What were your goals for enrolling into the doctorate?".

To construct motivational profiles of doctoral students, we applied latent class analysis (LCA). LCA is a statistical technique that identifies an unobserved categorical variable based on patterns in observed categorical indicators. In our case, it provides a class (motivational profile), based on the presence or absence of various enrollment motivations. In a sense, it resembles factor analysis, but is specifically designed for categorical rather than continuous variables. This method enables the identification of coherent motivational profiles, which is more appropriate than analyzing motivations separately and aligns with previous studies [49,59,60]. Based on goodness-of-fit statistics and interpretability, we selected a four-class model (see S2 Data for details and visualisation). The labels for all classes were chosen by the authors based on item-response probabilities. The probability of selecting each motive for each class is illustrated in Fig 2.

The most widespread class is labeled "Academic Orientation" (41% of respondents). Members of this class do not exhibit non-academic motivation (e.g., career advancement outside the university – 8%, military draft deferment – 3%, access to dormitory housing – 0%, etc.). This group represents typical PhD applicants within the Humboldtian model and includes those most oriented towards pursuing an academic career (47%). According to self-determination theory [47,48], this corresponds with intrinsic motivation, which is associated with a genuine interest in studying and working in science.

| Class | To receive PhD degree | To receive the diploma without defending the thesis | To develop research skills | To develop teaching skills | To continue research in the field of interest | To move forward in academic career | To move forward in non-academic career | To get an opportunity to travel abroad | To employ in the university / scientific organization | To get the deferment from the army | To get a place in a dormitory |
|---|---|---|---|---|---|---|---|---|---|---|---|
| Academic orientation (41%) | 96 | 4 | 69 | 72 | 37 | 47 | 11 | 8 | 37 | 3 | 0 |
| Unconscious motives (32%) | 69 | 19 | 26 | 10 | 15 | 12 | 21 | 3 | 5 | 24 | 5 |
| Topic devoted (14%) | 89 | 0 | 100 | 34 | 94 | 29 | 13 | 11 | 10 | 13 | 8 |
| Everything Everywhere All at Once (13%) | 99 | 9 | 100 | 78 | 82 | 100 | 38 | 46 | 72 | 22 | 10 |

**Fig 2. Motivational patterns based on LCA.** The numbers in the cells show conditional probability of selecting a given motive for each class.

The second class, "Unconscious Motives," accounts for 32% of the sample and is characterized by the lowest prevalence of enrollment motives. Apart from the most common motive across all groups – "to receive a PhD degree" – all other motives are expressed by no more than a quarter of respondents. Additionally, non-academic goals, such as obtaining a degree without defending a dissertation (19%) or deferring military service (24%), are most common in this class. This group includes PhD students who lacked clear goals at the time of enrollment and constitutes approximately one-third of all respondents. In terms of self-determination theory [47,48], these students exhibit amotivational attitudes, as they have no clear goals, or their goals remain unarticulated. This is also reflected in the way their goals are formulated – focusing on avoiding undesirable outcomes rather than pursuing positive ones (e.g., avoiding mandatory military service).

The third class, "Topic Devoted," comprises 14% of the sample. These individuals are less focused on teaching skills (34%) or career development outside academia (13%) but demonstrate a strong orientation toward researching a specific topic (94%) and developing research skills (100%). We assume that, for this group, pursuing a PhD is primarily driven by a deep personal interest in their potential dissertation research, which also corresponds to intrinsic motivation [47,48].

Finally, the "Everything Everywhere All at Once" class accounts for 13% of the sample. This class is characterized by a broad motivational profile, encompassing nearly all goals – from securing employment at a university (72%) to advancing in a non-academic career (38%). Members of this group are also interested in developing research (100%) and teaching skills (78%), as well as exploring opportunities for academic mobility (46%) more often than members of the other classes. We hypothesize that individuals in this class anticipate a highly diverse and enriched PhD experience. This aligns with integrated motivation – a blend of intrinsic and extrinsic motivation, where the locus of control is external, yet the activity remains engaging and meaningful to the individual [47,48].

## Portrait of groups

Next, we examine the relationship between the identified motivational classes and individual characteristics of the participants in order to develop more detailed profiles. Since all analyzed variables are categorical, we used chi-squared tests to assess the significance of group differences. Fig 3 presents the distribution of individual characteristics across classes, along with corresponding significance levels.

We observe significant differences in motivational profiles based on gender. Men are much more likely to exhibit unconscious motives, while women predominantly demonstrate academic orientations. This discrepancy can largely be attributed to mandatory military service for men in Russia, as enrolling in a PhD program provides an opportunity to defer conscription.

A somewhat paradoxical finding is that part-time PhD students are more likely to exhibit academic orientations compared to their full-time counterparts, who are more frequently associated with the "Topic Devoted" and "Everything Everywhere All at Once" classes. We hypothesize that applicants with these motivational profiles tend to choose full-time programs, recognizing that full-time study offers a richer PhD experience and greater opportunities to delve deeply into a specific topic. This hypothesis is further supported by differences in tuition payment structures: those seeking an enriched experience or focused research are more often enrolled tuition-free, while fee-paying PhD students are more frequently characterized by unconscious motives. This trend is also reflected in the analysis by university type. Leading universities have significantly fewer students with non-academic or unconscious motives, likely due to more holistic selection procedures that assess not only formal criteria but also applicants' motivations and commitment to academic pursuits [40].

Educational trajectories are also linked to motivational profiles. Returners — students entering a PhD program after a gap of five years or more [64,65] — are more likely to belong to the "Academic Orientation" class. In Russia, a significant gap before entering a PhD program is common among individuals already employed at a university, mostly in teaching positions, who pursue a degree primarily for career advancement in the academic sector [7].

Following Bekova's classification [62], we divided PhD students into four categories based on their employment status and examined the relationship with motivational profiles. As expected, unconscious motives are more prevalent among

| Profile characteristic | Academic orientation | Unconscious motives | Topic devoted | Everything Everywhere All at Once | $\chi 2$ |
|---|---|---|---|---|---|
| **Gender** | | | | | |
| Male | 31 | 38 | 16 | 15 | 48,97*** |
| Female | 50 | 26 | 12 | 12 | |
| **Field of study** | | | | | |
| Humanities | 46 | 28 | 11 | 15 | 21,51* |
| Social sciences | 43 | 32 | 11 | 14 | |
| Math and Earth Sciences | 38 | 32 | 20 | 10 | |
| Engineereing and Technical Sciences | 37 | 35 | 14 | 13 | |
| **Form of studying** | | | | | |
| Full-time | 39 | 32 | 14 | 15 | 19,79*** |
| Part-time | 53 | 32 | 9 | 6 | |
| **Form of financing** | | | | | |
| Tuition-free | 40 | 29 | 15 | 15 | 25,08*** |
| Tuition-based | 45 | 39 | 9 | 7 | |
| **Leading university** | | | | | |
| Yes | 39 | 25 | 17 | 20 | 50,17*** |
| No | 43 | 37 | 11 | 9 | |
| **Trajectory** | | | | | |
| Direct pathway student | 41 | 36 | 13 | 10 | 48,03*** |
| Interrupter | 41 | 23 | 17 | 19 | |
| Returner | 47 | 38 | 8 | 7 | |
| **Employment status** | | | | | |
| Not employed | 40 | 26 | 12 | 22 | 48,91*** |
| Employment outside university | 36 | 39 | 14 | 11 | |
| Research position at university | 49 | 21 | 14 | 16 | |
| Another position at university | 47 | 31 | 13 | 10 | |
| **Majored at the same field before** | | | | | |
| Yes | 42 | 29 | 14 | 14 | 21,38*** |
| No | 35 | 47 | 9 | 9 | |
| **Plans to proceed with academic career** | | | | | |
| Yes | 47 | 22 | 15 | 16 | 243,26*** |
| No | 18 | 74 | 6 | 1 | |
| **Total sample** | | | | | |
| Share of class | 41 | 32 | 14 | 13 | |

*** $p < 0.001$, ** $p < 0.01$, * $p < 0.05$

**Fig 3. Motivational profiles by socio-demographic and educational characteristics.** Percentages of respondents within each class; chi-squared values for significance of group differences.

those employed outside academia, whereas students working in universities — whether in research or other positions — demonstrate academic motivations.

Finally, differences in motivational profiles are observed between those who changed their field of study and those who remained within the same discipline. Changing fields is more common among individuals with unclear motives for enrollment, suggesting that their decision to pursue a PhD was less deliberate. Conversely, those planning to focus on a specific research topic are more likely to remain within their original field of study.

### Motives, trajectories, and doctoral experience

Next, we use the identified classes as predictors in a series of regression models to test the relationship between motivational profiles and various doctoral outcomes – satisfaction with the doctorate, self-assessed confidence in dissertation defense, and plans to pursue research career. All regressions include the same set of control variables, comprising standard sociodemographic indicators (gender, marital status) and several specific control factors previously shown to be significant in studies of Russian doctoral students. These controls include employment status [62], university type, and year of study [37].

### Motivations and satisfaction

The first model concerns satisfaction with doctoral studies. It was measured using the question: "*Please, estimate how satisfied you are with your doctoral studies in general?*" (Scale: 1 — "Not satisfied at all", 2 — "Rather not satisfied", 3 — "Rather satisfied", 4 — "Totally satisfied"). A binary satisfaction variable was created, where responses of 1 and 2 are coded as 0, and responses of 3 and 4 are coded as 1.

Results of binomial logistic regression are presented in Table 1. We find that motivational profile is a significant predictor of overall satisfaction with the doctoral experience. Compared to the reference category (Academic Orientation), members of the "Unconscious Motives" class are nearly half as likely to be satisfied with their education, all other things being equal ($p < 0.001$). Differences between the other classes and the reference category are not statistically significant. Therefore, only differences between the "Academic Orientation" and "Unconscious Motivation" groups are interpreted.

### Motivations and self-assessed confidence

The second model examines motivational profiles in relation to doctoral students' self-assessed likelihood of successfully defending their dissertation. Following the approach proposed by Zhuchkova et al. [66], we constructed a variable called "lack of confidence" using categorical principal component analysis (CatPCA), which allows for the extraction of a continuous component from categorical indicators. The questionnaire items used, along with component loadings and communalities, are presented in Table 2.

These items demonstrate a high level of consistency (Cronbach's alpha = 0.89). The resulting index represents doctoral students' unconfidence in successfully completing their program and defending their dissertation, with higher values indicating lower confidence.

We now use it as a dependent variable in a linear regression, its output is presented in Table 3. There is a significant association between motivational profile and doctoral students' lack of confidence in successfully defending their dissertation. Students in the "Unconscious Motives" class exhibit higher lack of confidence compared to the reference category (Academic Orientation). Once again, the other classes do not show statistically significant differences from the reference category and therefore are not interpreted.

### Motivations and career aspirations

The third model tests the association between motivational profiles and plans to pursue a research career after completing the PhD. To measure career aspirations, we used the following single-choice question, with the share of participants selecting each option indicated in parentheses:

**Table 1. Results of the first model (DV — satisfaction with the doctorate).**

| Variable | β | SE | Odds ratio | LB CI 97.5% | UB CI 97.5% |
|---|---|---|---|---|---|
| Intercept | 1.63*** | 0.35 | 5.10 | 2.57 | 10.34 |
| Class (reference category – Academic orientation) | | | | | |
| Unconscious Motives | −0.77*** | 0.19 | 0.46 | 0.32 | 0.67 |
| Topic Devoted | −0.25 | 0.25 | 0.78 | 0.49 | 1.28 |
| Everything Everywhere All at Once | −0.19 | 0.24 | 0.82 | 0.52 | 1.33 |
| Employment status (reference category – Unemployed) | | | | | |
| Employed outside the university | 0.33 | 0.24 | 1.40 | 0.86 | 2.24 |
| Employed in a research position | 0.57* | 0.27 | 1.76 | 1.03 | 2.99 |
| Employed in other position | 0.53 | 0.31 | 1.70 | 0.93 | 3.12 |
| Female | 0.37* | 0.16 | 1.44 | 1.05 | 1.99 |
| Single | −0.05 | 0.16 | 0.95 | 0.69 | 1.30 |
| Part-time | −0.03 | 0.34 | 0.97 | 0.49 | 1.91 |
| Tuition-based | 0.50 | 0.28 | 1.64 | 0.96 | 2.91 |
| Leading university | −0.64*** | 0.17 | 0.53 | 0.38 | 0.74 |
| Field of study (reference category – Math and Earth sciences) | | | | | |
| Engineering and technical sciences | −0.20 | 0.24 | 0.82 | 0.51 | 1.32 |
| Social sciences | −0.59* | 0.23 | 0.56 | 0.35 | 0.87 |
| Humanities | −0.28 | 0.23 | 0.75 | 0.48 | 1.18 |
| Year of study (reference category – 1) | | | | | |
| 2 | 0.09 | 0.18 | 1.09 | 0.77 | 1.56 |
| 3 | 0.23 | 0.21 | 1.25 | 0.84 | 1.89 |
| 4 | 0.33 | 0.32 | 1.39 | 0.76 | 2.65 |

Note:

*$p < 0.05$;

**$p < 0.01$;

***$p < 0.001$; Nagelkerke's $R^2$ = 0.1; AIC = 1,152; N = 1,219.

**Table 2. Component loadings and communalities of catPCA model – lack of confidence.**

| Variable (I am afraid…) | Loading | Communality |
|---|---|---|
| I will not defend my dissertation in time | 0.852 | 0.726 |
| I will not pass my next examination | 0.841 | 0.708 |
| All my work is being done in vain and does not get me closer to the defense | 0.913 | 0.833 |
| That my research does not correspond to the level of a doctoral dissertation | 0.871 | 0.759 |
| I am not capable of completing my dissertation | 0.915 | 0.837 |
| Eigenvalues | 3.86 | |
| Variance accounted for (%) | 77.3 | |

*Tell us about your career plans after graduation. Do you plan to keep conducting research after your thesis defense?*

1. *Yes, I do (82%)*

2. *No, I do not (18%)*

**Table 3. Results of the second model (DV — lack of confidence in dissertation defense).**

| Variable | β | SE | LB CI 97.5% | UB CI 97.5% |
|---|---|---|---|---|
| Intercept | −0.43** | 0.14 | −0.70 | −0.16 |
| Class (reference category – Academic orientation) | | | | |
| Unconscious Motives | 0.29*** | 0.07 | 0.15 | 0.43 |
| Topic Devoted | −0.06 | 0.09 | −0.24 | 0.12 |
| Everything Everywhere All at Once | 0.10 | 0.09 | −0.07 | 0.28 |
| Employment status (reference category – Unemployed) | | | | |
| Employed outside the university | −0.01 | 0.10 | −0.21 | 0.18 |
| Employed in a research position | −0.20 | 0.11 | −0.41 | 0.01 |
| Employed in other position | −0.01 | 0.12 | −0.25 | 0.22 |
| Female | 0.18** | 0.06 | 0.06 | 0.30 |
| Single | 0.12* | 0.06 | 0.00 | 0.24 |
| Part-time | −0.07 | 0.12 | −0.30 | 0.16 |
| Tuition-based | −0.10 | 0.10 | −0.31 | 0.10 |
| Leading university | 0.19** | 0.06 | 0.06 | 0.31 |
| Field of study (reference category – Math and Earth sciences) | | | | |
| Engineering and technical sciences | 0.15 | 0.09 | −0.02 | 0.33 |
| Social sciences | 0.09 | 0.09 | −0.08 | 0.26 |
| Humanities | 0.07 | 0.09 | −0.09 | 0.24 |
| Year of study (reference category – 1) | | | | |
| 2 | 0.18** | 0.07 | 0.04 | 0.32 |
| 3 | 0.13 | 0.08 | −0.02 | 0.29 |
| 4 | 0.01 | 0.11 | −0.20 | 0.22 |

**Note**:

*$p < 0.05$;

**$p < 0.01$;

***$p < 0.001$; = 0.06; F = 3.81; p-value < 0.001; N = 1,071.

The output is presented in Table 4. The model indicates that motivational profiles are significantly associated with plans to pursue a research career. Two classes stand out in comparison to the reference category (Academic Orientation). Respondents characterized by unconscious motives are significantly less likely to plan for a research career, whereas those in the "Everything Everywhere All at Once" class are more likely to choose it in the future.

### Can experience reshape motivation?

Finally, we examine the subsample of individuals in the "Unconscious Motives" class to analyze which elements of the doctoral experience are associated with the development of research career aspirations. The fourth model focuses exclusively on the subsample of the "Unconscious Motives" class and incorporates components of the doctoral experience as covariates. This model examines which aspects of the doctoral experience help students with non-academic or unclear motives develop an intention to pursue a research career.

To measure different aspects of doctoral experience, we used the following single-choice question: "*To what extent are the following aspects present during your doctoral studies?*" (Scale: 3 – "To a great extent", 2 – "To some extent", 1 – "Not at all present"). The question included 19 statements. For subsequent analysis, these statements were grouped into several components using categorical principal component analysis (catPCA). The analysis was conducted on a polychoric

**Table 4. Results of the third model (DV — plans to pursue a research career).**

| Variable | β | SE | Odds ratio | LB CI 97.5% | UB CI 97.5% |
|---|---|---|---|---|---|
| Intercept | 2.34*** | 0.41 | 10.33 | 4.74 | 23.40 |
| Class (reference category – Academic orientation) | | | | | |
| Unconscious Motives | −2.00*** | 0.20 | 0.14 | 0.09 | 0.20 |
| Topic Devoted | 0.03 | 0.32 | 1.03 | 0.56 | 1.98 |
| Everything Everywhere All at Once | 1.66** | 0.61 | 5.27 | 1.87 | 22.08 |
| Employment status (reference category – Unemployed) | | | | | |
| Employed outside the university | −0.38 | 0.30 | 0.68 | 0.37 | 1.21 |
| Employed in a research position | 0.67 | 0.35 | 1.95 | 0.97 | 3.86 |
| Employed in other position | 0.05 | 0.36 | 1.05 | 0.51 | 2.13 |
| Female | 0.17 | 0.18 | 1.18 | 0.82 | 1.69 |
| Single | 0.02 | 0.17 | 1.02 | 0.73 | 1.44 |
| Part-time | 0.52 | 0.33 | 1.67 | 0.89 | 3.20 |
| Tuition-based | 0.15 | 0.27 | 1.16 | 0.69 | 1.97 |
| Leading university | 0.10 | 0.19 | 1.11 | 0.77 | 1.60 |
| Field of study (reference category – Math and Earth sciences) | | | | | |
| Engineering and technical sciences | −0.34 | 0.25 | 0.71 | 0.44 | 1.15 |
| Social sciences | −0.18 | 0.25 | 0.83 | 0.50 | 1.37 |
| Humanities | −0.04 | 0.25 | 0.96 | 0.58 | 1.58 |
| Year of study (reference category – 1) | | | | | |
| 2 | −0.01 | 0.21 | 0.99 | 0.66 | 1.49 |
| 3 | −0.18 | 0.22 | 0.83 | 0.54 | 1.28 |
| 4 | 0.13 | 0.32 | 1.14 | 0.61 | 2.16 |

**Note**:

*$p < 0.05$;

**$p < 0.01$;

***$p < 0.001$; Nagelkerke's $R^2$ = 0.32; AIC = 969; N = 1,263.

correlation matrix using the psych package in R. The promax rotation was applied, as the components of the doctoral experience were expected to be correlated.

To determine the optimal number of components, we compared several different models. A six-component model was ultimately selected due to its interpretability. It explains 78% of the variance in the original variables related to the doctoral experience. The variable "The need to use English or another foreign language" was excluded due to interpretation challenges. The resulting components capture distinct aspects of the doctoral experience:

1. *Interaction with department;*

2. *International activities;*

3. *Educational workload;*

4. *Interaction with peers;*

5. *Research activities;*

6. *Interaction with supervisor.*

Component loadings and communalities are presented in Table 5.

**Table 5. Component loadings and communalities of the catPCA model – aspects of doctoral experience.**

| Variable | RC1 | RC2 | RC3 | RC4 | RC5 | RC6 | Communality |
|---|---|---|---|---|---|---|---|
| Assistance and support from the supervisor | 0.16 | −0.05 | −0.06 | −0.02 | −0.07 | 0.91 | 0.89 |
| Regular communication with the supervisor | −0.08 | −0.04 | −0.08 | 0.06 | 0.02 | 0.97 | 0.90 |
| Regular monitoring of my progress by the supervisor | 0.20 | 0.05 | 0.06 | −0.14 | −0.05 | 0.88 | 0.88 |
| A lot of work on writing academic texts (articles. monographs. dissertations) | 0.15 | −0.20 | 0.06 | −0.19 | 0.93 | 0.01 | 0.80 |
| Attending scientific conferences | 0.19 | 0.01 | −0.10 | −0.14 | 0.91 | −0.23 | 0.68 |
| A lot of research work on data collection (experiments. field work) | −0.41 | 0.04 | 0.10 | 0.14 | 0.75 | 0.19 | 0.68 |
| International internships | 0.05 | 1.02 | 0.08 | −0.10 | −0.11 | −0.09 | 0.87 |
| Work in international research teams | 0.04 | 1.02 | 0.07 | −0.11 | −0.10 | 0.03 | 0.90 |
| Work on team projects of the scientific department | −0.08 | 0.53 | −0.08 | 0.17 | 0.26 | 0.12 | 0.67 |
| Regular communication with other doctoral students | −0.04 | −0.13 | 0.11 | 1.01 | −0.09 | 0.04 | 0.84 |
| Assistance and support from other doctoral students | 0.28 | −0.06 | 0.07 | 0.90 | −0.18 | −0.13 | 0.82 |
| Regular communication with other researchers at the university/ research institute | 0.20 | 0.13 | −0.15 | 0.39 | 0.35 | −0.04 | 0.68 |
| High requirements for educational and scientific results | 0.51 | −0.07 | 0.41 | −0.14 | 0.26 | 0.12 | 0.73 |
| Regular monitoring of my progress by other staff of the department/ doctoral school | 0.79 | 0.04 | 0.07 | 0.04 | −0.06 | 0.07 | 0.72 |
| Feedback on my research work from senior colleagues | 0.55 | 0.17 | −0.13 | 0.10 | 0.14 | 0.09 | 0.71 |
| Assistance and support from other university/research institute staff | 0.65 | −0.02 | −0.14 | 0.31 | 0.00 | 0.06 | 0.76 |
| A lot of stress associated with studying in doctoral school | −0.01 | 0.14 | 0.91 | 0.01 | −0.08 | −0.10 | 0.80 |
| A large educational load | 0.00 | 0.00 | 0.77 | 0.24 | 0.12 | 0.03 | 0.72 |

**Note:** RC1 stands for interaction with department, RC2 – international activities, RC3 – educational workload, RC4 – interaction with peers, RC5 – research activities, RC6 – interaction with supervisor.

Six variables were extracted – standardized factor scores of components, where higher values of these variables indicate a greater prominence of the corresponding aspects in students' experiences. We use these six variables to find out which aspects of doctoral experience are related to research-oriented career aspirations among those who initially enrolled with non-academic motives. See Table 6 for regression coefficients.

The only significant predictor in the model is engagement into research activities. This variable is positively associated with the intention to pursue a research career after completing the PhD ($p < 0.001$). Thus, more intensive involvement into academic writing, filed work and scientific conferences contribute to the development of research career trajectories, even among PhD students with unconscious motives.

## Conclusion and discussion

This study underscores the pivotal role of motivational profiles in shaping various doctoral outcomes, including satisfaction, academic self-esteem, and career aspirations. Our findings reveal that doctoral students with academic orientations consistently exhibit the most favorable outcomes, including higher satisfaction with their studies, greater confidence in successfully defending their dissertations, and a stronger likelihood of pursuing research careers. Conversely, students with unconscious motives face significant challenges, including lower satisfaction, heightened lack of confidence, and diminished intentions to pursue a research career. These differences highlight the impact of non-academic motivation, such as military service deferment or unclear goals at the time of enrollment, on doctoral outcomes.

On the one hand, our results indicate that 68% of Russian doctoral students report primarily academic motivations for enrollment. By comparison, recent studies show that academic motivations are typical for 61% of doctoral students in Latvia [18], while in South Korea, the average score for the variable "motivation for an academic profession" was 5.3 out of 7, indicating a high level of academic motivation among respondents [57].

**Table 6. Results of the fourth model (DV — plans to pursue research career, subsample).**

| Variable | β | SE | Odds ratio | LB CI 97.5% | UB CI 97.5% |
|---|---|---|---|---|---|
| Intercept | 0.87 | 0.52 | 2.38 | 0.86 | 6.76 |
| Interaction with supervisor | −0.12 | 0.13 | 0.88 | 0.68 | 1.14 |
| Research activities | 0.73*** | 0.16 | 02.07 | 1.54 | 2.83 |
| International activities | 0.21 | 0.16 | 1.23 | 0.91 | 1.70 |
| Interaction with peers | −0.02 | 0.15 | 0.98 | 0.74 | 1.31 |
| Interaction with department | 0.13 | 0.15 | 1.14 | 0.86 | 1.52 |
| Educational workload | −0.15 | 0.12 | 0.86 | 0.68 | 01.09 |
| Employment status (reference category – Unemployed) | | | | | |
| Employed outside the university | −0.49 | 0.42 | 0.61 | 0.27 | 1.38 |
| Employed in a research position | 0.87 | 0.51 | 2.39 | 0.87 | 6.64 |
| Employed in other position | 0.13 | 0.50 | 1.13 | 0.42 | 03.03 |
| Female | 0.15 | 0.27 | 1.16 | 0.68 | 1.95 |
| Single | −0.04 | 0.24 | 0.97 | 0.60 | 1.55 |
| Part-time | 0.42 | 0.42 | 1.53 | 0.67 | 3.54 |
| Tuition-based | 0.26 | 0.34 | 1.30 | 0.67 | 2.54 |
| Leading university | 0.40 | 0.27 | 1.50 | 0.89 | 2.53 |
| Field of study (reference category – Math and Earth sciences) | | | | | |
| Engineering and technical sciences | −0.36 | 0.34 | 0.70 | 0.36 | 1.35 |
| Social sciences | −0.34 | 0.35 | 0.71 | 0.36 | 1.42 |
| Humanities | −0.20 | 0.36 | 0.82 | 0.40 | 1.65 |
| Year of study (reference category – 1) | | | | | |
| 2 | −0.48 | 0.29 | 0.62 | 0.35 | 1.10 |
| 3 | −0.46 | 0.30 | 0.63 | 0.35 | 1.15 |
| 4 | −0.34 | 0.44 | 0.71 | 0.30 | 1.72 |

**Note**:

*$p < 0.05$;

**$p < 0.01$;

***$p < 0.001$; Nagelkerke's $R^2 = 0.26$; AIC $= 503$; N $= 399$.

However, a more detailed breakdown of our results reveals several concerning issues. First, there is evidence of a Matthew effect: leading universities attract the most motivated doctoral students, thereby exacerbating gaps and educational inequality. This is supported by our analysis in the "Portrait of groups" section (see Fig 3), which shows that students from leading universities are significantly less likely to belong to the "Unconscious Motives" class — a profile associated with unclear or non-academic enrollment goals, such as deferring military service. The overconcentration of doctoral education in leading universities in Russia [37], combined with unclear motives of doctoral students in non-elite universities, presents a significant challenge for regional universities in terms of (re)producing human capital.

Second, employment in non-academic sectors is more frequently associated with unconscious motives, and the practice of combining work and study is widespread in Russia [63]. This presents a substantial risk: doctoral students may be unable to fully immerse themselves in the university environment, making it more difficult to develop intrinsic motivation to enhance persistence and achievement [47,49]. This challenge, driven by the necessity of balancing work and study, reignites the discussion about the insufficient funding of Russian doctoral education [23,67]. While our findings are grounded in the context of Russian doctoral education, similar patterns may be observed in other systems that have experienced an expansion of non-traditional or professionally oriented doctoral pathways. For example, online EdD programs in countries

like the United States often enroll candidates with diverse academic and career backgrounds [68], whose initial motivations may not align with traditional research-oriented goals. Although we do not explore these models in depth, the parallels suggest that the challenges associated with heterogeneous motivations are not unique to Russia and may warrant further comparative research.

Another key finding is that engagement in research activities is the only factor within the doctoral experience that fosters the development of a research career aspirations among those who initially enrolled with unconscious motives. Previous research has consistently emphasized the importance of academic engagement in the doctorate [69], showing that it supports skills development [70], identity formation [71], and the enhancement of student wellbeing [4]. However, survey data from Russia reveal that 40% of doctoral students never participated in a research project beyond their thesis, and only 13% reported involvement in research projects outside their university [59]. It can be hypothesized that doctoral students with non-academic motivations might feel excluded from academia due to assumptions about its elite and purely scientific axiology [55]. Yet, if they are given the opportunity to participate in research activities, even those who initially lack motivation may internalize academic norms and eventually develop research-oriented career aspirations. In terms of Self-Determination Theory [47,48], this can be interpreted as a shift from external to internal motivation, fostered by the satisfaction of the needs for competence, autonomy, and relatedness through research activities.A major limitation of this study is its reliance on retrospective self-reports of motivations for entering doctoral programs. Although the year of study was included as a control variable in the regressions, the survey data may not fully capture students' motivations at the time of admission, as respondents are likely reconstructing their motivations retrospectively. The same limitation applies to self-reported intentions regarding thesis defense and research career plans, as prior studies have demonstrated that these indicators are not entirely reliable measures of educational outcomes [72]. Additionally, career aspirations were measured at different stages of doctoral education rather than exclusively in the final year. It is likely that some respondents changed their career aspirations over time. Nevertheless, despite these limitations, the findings reveal clear patterns of motivational profiles. Notably, one in three doctoral students exhibits amotivational tendencies, and non-academic motivations for pursuing doctoral studies remain relatively common.

The favorable position of academically oriented doctoral students may indicate a misalignment between the institutional structure of Russian doctoral education, which remains largely focused on academic careers, and the increasingly complex and diverse expectations of applicants. Several structural challenges shape this misalignment, including rigid admission procedures [40], a lack of program differentiation [23], and the concentration of doctoral students in leading universities [73]. These factors directly impact the educational experience of students with varying motivations. One possible interpretation is that unmotivated doctoral students made a mistake in selecting their educational trajectory and that doctoral education should filter them out to prevent an educational mismatch. However, given the increasingly heterogeneous student body [7] and the evolving role of doctoral education [1], a more effective approach would involve diversifying doctoral programs, implementing targeted efforts to assess and address applicants' motivations at the point of entry, and providing institutional support for the academic integration of students who initially do not demonstrate strong research motivation. By addressing these structural barriers, Russian doctoral education can better align with the diverse needs of students, ultimately ensuring greater academic and professional success for doctoral candidates across all motivational profiles.

## Supporting information

**S1 Data. Anonymized data used in the research.**
(XLSX)

**S2 Data. Goodness-of-fit statistics for latent class analysis models.**
(DOCX)

## Author contributions

**Conceptualization:** Nikita Smirnov, Elena Tarasova.

**Formal analysis:** Nikita Smirnov.

**Methodology:** Nikita Smirnov.

**Project administration:** Nikita Smirnov, Elena Tarasova.

**Supervision:** Nikita Smirnov.

**Writing – original draft:** Nikita Smirnov, Elena Tarasova.

**Writing – review & editing:** Nikita Smirnov, Elena Tarasova.

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
