## [Decision Letter · Decision Letter 0]

15 Apr 2025

PONE-D-25-07628Academic or accidental? How enrollment motives shape doctoral experiences and career aspirationsPLOS ONE

Dear Dr. Smirnov,

Thank you for submitting your manuscript to PLOS ONE. After careful consideration, we feel that it has merit but does not fully meet PLOS ONE’s publication criteria as it currently stands. Therefore, we invite you to submit a revised version of the manuscript that addresses the points raised during the review process.

You have several different reviews here. Two from reviewers and one that I added in the earlier section. The manuscript has interesting ideas and I like the notion of an LCA to show different groups who want to get a doctorateThe main problem I have with things is the organization and the lack of information about both the data analysis and the procedure. It is difficult to review given the flow of information. I did include a video of how to report logistic regressions and I am not sure why you did 6 separate ones when you could do a multiple logistic regression. So please rethink your organization and provide more details about the various issues.

We look forward to receiving your revised manuscript.

Kind regards,

Mary Diane Clark, PhD

Academic Editor

PLOS ONE

Journal Requirements:

Additional Editor Comments:

Thank you for submitting you manuscript to PLOS ONE. We have two reviews and I have also completed a review on the manuscript. I believe it needs a major re-write to make it more readable. So let me go through my concerns here for you.

1--the literature review is redundant. You have the three sections and repeat much of the information from one section in another one. Please combine all three into one section.

2--Your participant section needs to have more definition of your sample.

3--you need a procedure section with materials that are more defined.

4--You need to have a highly well define Data Analysis section. Statistics come without the reader being prepared. Many are not listed in results but in other sections. You could report the data as being conducted in phases---One the LCA, then the regressions etc. However, you need to better set up each of those statistical methods In the LCA you need to have the traditional graphs of how much variance is contributed by each factor and maybe even a scree plot. Also, it would be nice to set the 4 categories with each of their predictors in a model using --- you need more visual on your outcomes and more information on how you set up the analysis

4A--I have included a link here https://www.youtube.com/watch?v=VjjrM7WMPh4

on how to report logistic regressions ----

the way it is set up is confusing

Also not sure why you did different ones rather than one multiple regression which might show how the variables interact.

It would be helpful to have the specificity and sensitivity of the classifications for each of your 4 latent classes

Figure 1 is not clear enugh for publication

Other Figures need clear title and explanations.

The profiles section is a bit confusing and how ou describe the models also. Again set up the data analytic section and describe each one. Then present the results for each ---most likely as separate phases of the project

Reviewers' comments:

Reviewer's Responses to Questions

**Comments to the Author**

1. Is the manuscript technically sound, and do the data support the conclusions?

Reviewer #1: Yes

Reviewer #2: Partly

2. Has the statistical analysis been performed appropriately and rigorously? 

Reviewer #1: Yes

Reviewer #2: I Don't Know

3. Have the authors made all data underlying the findings in their manuscript fully available?

Reviewer #1: Yes

Reviewer #2: Yes

4. Is the manuscript presented in an intelligible fashion and written in standard English?

Reviewer #1: No

Reviewer #2: Yes

5. Review Comments to the Author

Reviewer #1: I found the article to be an interesting read, as it provided valuable insights into PhD programs in Russia and the experiences of students on their doctoral journeys.

Here are several editing suggestions:

1) Fully expand the term "OECD" before using its acronym (line 54).

2) In Table 1, use periods instead of commas for decimal values (e.g., 1.59 instead of 1,59) (line 360).

3) In Table 2, use periods instead of commas (line 371).

4) Combine lines 372 and 373 into a single line.

5) In Table 3, use periods instead of commas (line 383).

6) In Table 4, use periods instead of commas (line 393).

7) In the References section (lines 471-689), correct the formatting to align with the required style.

8) In Figure 1, the text is difficult to read; consider changing the color to something other than red.

9) In Figure 2, differentiate each column for easier reading.

10) In Figure 3, differentiate each column as well for clarity.

11) In Appendix 1 (S2), use periods instead of commas.

12) In Appendix 2 (S3), use periods instead of commas.

Reviewer #2: Reviewer Comments:

Lines 215–216: The sequence of questions—beginning with an open-ended item and immediately followed by a closed-ended one—limits the potential richness of responses. The second question (“Do you plan to keep conducting research after your thesis defense?”) may have prematurely narrowed participant interpretation of the initial prompt (“Tell us about your career plans after graduation”).

Line 227: The section on “Academic Orientation” lacks clarity regarding participant distribution. Percentages or frequencies of respondents identifying with this orientation should be provided to enhance transparency and context.

Lines 282–283: The authors conflate academic and administrative career paths under the umbrella of “Academic Orientation.” This is problematic, as these trajectories are distinctly different—academic careers emphasize research and teaching, whereas administrative roles focus on institutional management and operations.

Title: The phrase “Academic or Accidental” is potentially problematic. The term accidental may carry a negative or dismissive connotation, implying that participants not aligned with an academic trajectory were misguided in pursuing doctoral studies. The framing risks marginalizing non-academic career intentions.

Participant Demographics: Key demographic variables such as age, marital status, and current income are not addressed. These factors could meaningfully influence doctoral students' motivations and career decision-making. For instance, older students or those with greater life responsibilities may be influenced by different factors than their younger counterparts.

Career Field Changes: The survey does not appear to explore whether participants had previously changed career fields or disciplines. Including such items could offer valuable insight into motivations for entering doctoral programs.

Clarity on “Doctoral Outcomes”: The stated aim, per the abstract, is to examine “doctoral outcomes.” However, the discussion reveals that data were collected from participants at varying stages of their doctoral journeys. This discrepancy is misleading, as “outcomes” typically implies post-doctoral experiences and results.

Institutional Context Missing: The discussion suggests that top-tier universities tend to admit academically oriented students, whereas lower-tier institutions are more open to diverse motivations. However, there is no information about the institutional affiliations of participants. Such data could meaningfully contextualize findings.

Underdeveloped Categories: The manuscript focuses primarily on comparisons between the Academic Orientation and Unconscious Motivation groups, with minimal attention to the Topic Devoted and Everything Everywhere All at Once categories. Even if findings from these groups were limited, this should be acknowledged. Revisiting and elaborating on these categories could yield additional insight and improve the overall interpretation of the data. As written, the treatment of these groups appears incomplete and secondary.

Overall Impression:

This manuscript offers an important and timely exploration of doctoral student motivations and career intentions. With further clarification of terminology, greater demographic detail, and more balanced treatment of all participant categories, this study has the potential to make a strong contribution to the field. The foundational work is promising, and the authors are encouraged to continue refining their analysis and framing for maximum impact.

6. PLOS authors have the option to publish the peer review history of their article (what does this mean? ). If published, this will include your full peer review and any attached files.

**Do you want your identity to be public for this peer review?** For information about this choice, including consent withdrawal, please see our Privacy Policy .

Reviewer #1: No

Reviewer #2: **Yes: ** Douglas C. Williams, Jr.

---

## [Author Response · Author response to Decision Letter 1]

7 May 2025

Dear Editor, Dr. Clark,

Dear Reviewers,

We appreciate your thoughtful feedback, which has helped us strengthen the paper. We have carefully addressed each comment and implemented revisions to enhance the clarity, accessibility, and rigor of the study. In the following, we provide detailed responses, highlighting the key modifications made to the manuscript.

We hope that the revised version meets the journal’s expectations, and we look forward to your further consideration.

Academic Editor

1. The literature review is redundant. You have the three sections and repeat much of the information from one section in another one. Please combine all three into one section.

Authors: We have revised the section to remove repetition and improve its structure. We have retained all key references and arguments, while consolidating overlapping parts and clarifying the distinction between different types of motivation. We believe the revised version is now clearer and more concise. See lines 136-172 for updated literature review.

2. Your participant section needs to have more definition of your sample.

Authors: To address this issue, we have expanded the “Data” section to provide a more detailed description of the sample design. Specifically, we now clarify the time frame of data collection, the CAWI methodology used, the quota-based sampling strategy (stratified by region and organization type), and the mechanisms for participant recruitment and response monitoring. See lines 175-206 for updated sample description.

3. You need a procedure section with materials that are more defined.

Authors: We appreciate the editor’s suggestion and have restructured the manuscript to improve clarity and organization. Specifically, we now present the empirical section in three distinct parts:

1. Procedure and materials, which describes the data collection process in detail, including the timeline, sampling strategy, and recruitment methods;

2. Data analysis, which outlines the analytic strategy, including the stepwise structure of latent class analysis and regression modeling;

3. Results, which focuses exclusively on presenting empirical findings without methodological explanations.

In the revised Procedure and materials section, we also provide an expanded description of the survey implementation process, including how participation was coordinated through the national monitoring system, the role of institutional representatives, and mechanisms for quota control and data cleaning.

4. You need to have a highly well define Data Analysis section. Statistics come without the reader being prepared. Many are not listed in results but in other sections. You could report the data as being conducted in phases — One the LCA, then the regressions etc. However, you need to better set up each of those statistical methods.

Authors: We have restructured the “Data Analysis” section to improve clarity and transparency. The revised version presents the analytic approach step by step: first, we describe the identification of latent motivational profiles using latent class analysis (LCA); next, we explain how these profiles are used as predictors in a series of regression models.

We now explicitly introduce LCA as follows:

“LCA is a statistical technique that identifies an unobserved categorical variable (i.e., latent class) based on patterns in observed categorical indicators — in our case, the presence or absence of various enrollment motives. In a sense, it resembles factor analysis but is specifically designed for categorical rather than continuous variables” (lines 222-225).

We have also added a new figure to S2 Appendix 1, where we visualize model fit statistics (AIC and BIC) to justify the selection of the four-class solution:

“The four-class solution demonstrates a joint minimum of both AIC and BIC. Starting from the five-class model, BIC begins to increase, indicating a decline in model parsimony. Based on this fit pattern, as well as considerations of interpretability, we selected the four-class model for further analysis”.

We then proceed with the subsection “Portrait of groups”, where bivariate relationships between motivational profiles and student characteristics are examined using chi-square tests.

Finally, in the subsection “Motives, trajectories, and doctoral experience”, we describe the logic behind our regression modeling, in which latent class membership serves as a key independent variable. This modeling strategy allows us to compare motivational profiles in terms of their association with doctoral outcomes.

5. In the LCA you need to have the traditional graphs of how much variance is contributed by each factor and maybe even a scree plot.

Authors: We appreciate the editor’s suggestion. While explained variance and scree plots are not typically used in latent class analysis — as the method is based on categorical latent variables rather than continuous dimensions — we agree that model fit indices should be clearly visualized.

Therefore, we have included a new figure in S2 Appendix 1 that plots key fit statistics (AIC and BIC) across 1 to 5 latent class models. This visualization serves a similar function to a scree plot in factor analysis by helping identify the optimal number of classes. Based on this plot and interpretability considerations, we selected the four-class solution.

6. Also, it would be nice to set the 4 categories with each of their predictors in a model using --- you need more visual on your outcomes and more information on how you set up the analysis

Authors: We appreciate the editor’s interest in better understanding how the latent classes were constructed and visualized. To clarify, the four motivational profiles were identified using latent class analysis (LCA), which does not rely on regression predictors to define the classes. Instead, LCA estimates the probability of selecting each enrollment motive conditional on class membership, allowing us to interpret distinct motivational patterns. These item-response probabilities are visualized in Figure 2, where each cell presents the conditional probability of selecting a given motive within each class.

Additionally, Figure 3 provides a portrait of each latent class in terms of socio-demographic and educational characteristics. It displays the distribution of respondent characteristics across classes, along with chi-square statistics to indicate significant group differences.

Finally, we describe each profile in detail within the “Motivational profiles of doctoral students” and “Portrait of groups” subsections of the Data Analysis section. We hope this clarifies that the motivational classes are derived through probabilistic modeling, not regression, and are already thoroughly visualized and interpreted in the manuscript.

7. I have included a link here https://www.youtube.com/watch?v=VjjrM7WMPh4 on how to report logistic regressions — the way it is set up is confusing. Also not sure why you did different ones rather than one multiple regression which might show how the variables interact.

Authors: Thank you for the suggestion and for providing the reference video. To improve clarity and consistency with common reporting standards, we have updated the regression tables by replacing the label “Exp(β)” with the more familiar term “Odds Ratio”, as suggested.

We also clarify that our regression analyses include:

1. One linear regression model to predict the continuous lack of confidence index (derived using CatPCA);

2. Several logistic regression models to predict binary outcomes, including doctoral satisfaction and intention to pursue a research career.

Each model includes multiple predictors, including motivational class membership and relevant demographic and institutional controls. We opted to estimate separate models for each outcome, as they are conceptually distinct and measured on different scales. Combining them in a single model would not be statistically appropriate.

All tables include full information: regression coefficients, standard errors, odds ratios (where applicable), 97.5% confidence intervals, p-values, and reference categories for categorical predictors. We hope this clarification addresses the reviewer’s concern and improves the accessibility of our statistical reporting. Should the editor or reviewers require additional statistical details or alternative formats, we would be happy to provide them.

8. It would be helpful to have the specificity and sensitivity of the classifications for each of your 4 latent classes

Authors: We appreciate the editor’s suggestion. However, we note that latent class analysis is not a supervised classification technique and does not involve ground-truth class labels (as is in machine learning research). Therefore, standard classification metrics such as specificity and sensitivity are not applicable in this context.

Instead, model quality in LCA is typically assessed through fit statistics (e.g., AIC, BIC etc.), classification entropy, and average posterior probabilities for class membership. In the revised manuscript, we report AIC and BIC for model selection (Table A1, Figure A1 in S2 Appendix 1).

However, to address this point, we have added Table A2 in S2 Appendix 1, which reports the average posterior probabilities of class membership by assigned class. This table demonstrates high diagonal values and low off-diagonal values, indicating strong class separation and reliable classification quality.

We hope this clarification addresses the editor’s concern regarding the robustness of class assignment.

9. Figure 1 is not clear enough for publication

Authors: Thank you for your feedback! We agree that the original version of Figure 1 required improvement. In the revised manuscript, we replaced Figure 1 with a new version that presents the overall distribution of enrollment motives using a high-resolution horizontal barplot. This version enhances clarity through the use of consistent formatting, legible labeling, and publication-quality resolution (600 dpi, TIFF-format).

10. Other Figures need clear titles and explanations.

Authors: We have revised the titles and captions of Figures 1-3 to make them more informative and self-explanatory. Specifically, we provided clearer titles that reflect the content and purpose of each figure, and added the captions to briefly explain the meaning of the values presented.

11. The profiles section is a bit confusing and how you describe the models also. Again set up the data analytic section and describe each one. Then present the results for each — most likely as separate phases of the project

Authors: To improve clarity, we enriched the profile descriptions with specific percentages from Figure 2 (lines 232-264).

Reviewer 1

1. Fully expand the term “OECD” before using its acronym (line 54).

Authors: We have clarified the abbreviation in the text: “...OECD (Organisation for Economic Co-operation and Development) countries…” (lines 54-55).

2. In Table 1, use periods instead of commas for decimal values (e.g., 1.59 instead of 1,59) (line 360).

Authors: We have replaced commas with periods in Table 1.

3. In Table 2, use periods instead of commas (line 371).

Authors: We have replaced commas with periods in Table 2.

4. Combine lines 372 and 373 into a single line.

Authors: We combined lines 372 and 373 into a single line.

5. In Table 3, use periods instead of commas (line 383).

Authors: We have replaced commas with periods in Table 3.

6. In Table 4, use periods instead of commas (line 393).

Authors: We have replaced commas with periods in Table 4.

7. In the References section (lines 471-689), correct the formatting to align with the required style.

Authors: We have revised the References section to ensure that all entries comply with the required formatting style, including the correct handling of long URLs and DOIs.

8. In Figure 1, the text is difficult to read; consider changing the color to something other than red.

Authors: Thank you for your feedback. We agree that the original version of Figure 1 required improvement. In the revised manuscript, we replaced Figure 1 with a new version that presents the overall distribution of enrollment motives using a high-resolution horizontal barplot. This version enhances clarity through the use of consistent formatting, legible labeling, and publication-quality resolution (600 dpi, TIFF-format).

9. In Figure 2, differentiate each column for easier reading.

Authors: We thank the reviewer for this observation! At this stage, we have retained the original visual style of Figure 2, as we believe it effectively communicates the differences in motivational profiles across latent classes in a compact and comparative format.

Nevertheless, we fully recognize the importance of clarity in visual presentation. To ensure that the figure meets all publication standards, we will consult the journal’s editorial team to confirm that the current design meets accessibility and formatting expectations. If any revisions are recommended by the editors at the production stage, we will promptly make the necessary adjustments.

10. In Figure 3, differentiate each column as well for clarity.

Authors: Thank you for this comment. We acknowledge the reviewer’s concern regarding the visual clarity of Figure 3. However, we have opted to retain the current design, as it provides a concise and structured overview of how socio-demographic and educational characteristics vary across motivational profiles. The format allows for direct comparisons across categories and highlights meaningful differences without overloading the figure.

That said, we will consult with the journal’s editorial team to ensure that the figure meets publication and accessibility standards. Should the editors recommend any modifications at the production stage, we will revise the figure accordingly.

11. In Appendix 1 (S2), use periods instead of commas.

Authors: We have replaced commas with periods in Appendix 1.

12. In Appendix 2 (S3), use periods instead of commas.

Authors: We have replaced commas with periods in Appendix 2.

Reviewer 2

1. Lines 215–216: The sequence of questions — beginning with an open-ended item and immediately followed by a closed-ended one — limits the potential richness of responses. The second question (“Do you plan to keep conducting research after your thesis defense?”) may have prematurely narrowed participant interpretation of the initial prompt (“Tell us about your career plans after graduation”).

Authors: The first phrase (“Tell us about your career plans after graduation”) served as an introduction to the question about academic career specifically. The narrowing of focus was intentional, as the primary aim was to assess participants’ plans for continuing research after their dissertation. We clarified in the text that the question refers specifically to plans for an academic career.

2. Line 227: The section on “Academic Orientation” lacks clarity regarding participant distribution. Percentages or frequencies of respondents identifying with this orientation should be provided to enhance transparency and context.

Authors: We have added the percentage of respondents for the “Academic Orientation” profile.

3. Lines 282–283: The authors conflate academic and administrative career paths under the umbrella of “Academic Orientation.” This is problematic, as these trajectories are distinctly different — academic careers emphasize research and teaching, whereas administrative roles focus on institutional management and operations.

Authors: We appreciate the reviewer’s observation and agree that academic and administrative career paths should be distinguished conceptually. Upon review, we recognized that our wording may have inadvertently blurred this distinction. In response, we have revised the relevant passage to more clearly reflect the focus on academic careers involving teaching and research, in line with the cited source. As is stated in Zhuchkova & Terentev, 2024:

For returners, it is more common to have a teaching position at their university: 70% of returners, 62% of interrupters, and 55% of DPS who work at their universities have a teaching position (χ2 = 23.47, p = 0.000).

So we altered the text of the manuscript:

In Russia, a signif

---

## [Decision Letter · Decision Letter 1]

2 Jun 2025

PONE-D-25-07628R1Motivation matters: how enrollment motives shape doctoral experiences and career aspirationsPLOS ONE

Dear Dr. Smirnov,

Thank you for submitting your manuscript to PLOS ONE. After careful consideration, we feel that it has merit but does not fully meet PLOS ONE’s publication criteria as it currently stands. Therefore, we invite you to submit a revised version of the manuscript that addresses the points raised during the review process. Thank you for your revisions. As you add the data it brings up some more issues.  I have attached a file to help you make the article clearer.  Please make the lit review more detailed and explain the theory that you seem to be using.

We look forward to receiving your revised manuscript.

Kind regards,

Mary Diane Clark, PhD

Academic Editor

PLOS ONE

Additional Editor Comments:

Thank you for your revision. As you expand on your analyses I have more questions. I have attached a file with my specific comments. I need to see a stronger lit review.

Then I need to see a well developed Data Analytic section that includes you questionnaire development and lays out all of the analysis that you do throughout the article

Then, the discussion needs to be stronger and broader. Honestly, I think that online and part time programs are very different from those at the top universities. Make your comments clearer in terms of define what the needs to different doctoral degrees are in the current environment. There are so many options ow and if you are arguing that we need to redesign doctoral degrees that would be important. The traditional research degree has changed and it would be good to have this article to start that conversation

However as written I am not seeing that clearly defined. Please try to combine the logistic regressions and make the flow of ideas clearer.

Reviewers' comments:

Reviewer's Responses to Questions

**Comments to the Author**

1. If the authors have adequately addressed your comments raised in a previous round of review and you feel that this manuscript is now acceptable for publication, you may indicate that here to bypass the “Comments to the Author” section, enter your conflict of interest statement in the “Confidential to Editor” section, and submit your "Accept" recommendation.

Reviewer #1: (No Response)

Reviewer #2: All comments have been addressed

2. Is the manuscript technically sound, and do the data support the conclusions?

Reviewer #1: Yes

Reviewer #2: Yes

3. Has the statistical analysis been performed appropriately and rigorously? 

Reviewer #1: Yes

Reviewer #2: I Don't Know

4. Have the authors made all data underlying the findings in their manuscript fully available?

Reviewer #1: Yes

Reviewer #2: Yes

5. Is the manuscript presented in an intelligible fashion and written in standard English?

Reviewer #1: Yes

Reviewer #2: Yes

6. Review Comments to the Author

Reviewer #1: The manuscript is generally well-written and scientifically sound. However, I recommend minor revisions to ensure consistency and accuracy in decimal formatting throughout the text, tables, and figures, including the Notes. There are inconsistencies in the use of decimal points versus decimal commas. Please standardize the format according to the journal's style guide.

Reviewer #2: Per the authors' responses to the reviewer's feedback, the manuscript is (in the reviewer's opinion) clear and appropriate for publication. The reviewer would like to congratulate the authors on their achievement and thank them for the detailed responses to each point of feedback.

7. PLOS authors have the option to publish the peer review history of their article (what does this mean? ). If published, this will include your full peer review and any attached files.

**Do you want your identity to be public for this peer review?** For information about this choice, including consent withdrawal, please see our Privacy Policy .

Reviewer #1: No

Reviewer #2: No

---

## [Author Response · Author response to Decision Letter 2]

9 Jul 2025

Response to Editor and Reviewers

Dear Editor, Dr. Clark,

Dear Reviewers,

We sincerely appreciate your deep engagement with the manuscript and your insightful comments. We hope that we managed to revise the text significantly in line with your suggestions. In an attached file, we respond to them one by one and the changes made in the text.

---

## [Decision Letter · Decision Letter 2]

31 Jul 2025

PONE-D-25-07628R2Motivation matters: how enrollment motives shape doctoral experiences and career aspirationsPLOS ONE

Dear Dr. Smirnov,

Thank you for submitting your manuscript to PLOS ONE. After careful consideration, we feel that it has merit but does not fully meet PLOS ONE’s publication criteria as it currently stands. Therefore, we invite you to submit a revised version of the manuscript that addresses the points raised during the review process. Reviewer 1 noted some requested changes that you had not made on your resubmission.  Can you please carefully review that review and respond as to how you have addressed those issues. They are minor and can easily be completed.  Then I will happily move the paper forward. 

We look forward to receiving your revised manuscript.

Kind regards,

Mary Diane Clark, PhD

Academic Editor

PLOS ONE

Journal Requirements:

Additional Editor Comments (if provided):

Can you please complete the requested changed from Reviewer 1? If you can do that we can move forward

Reviewers' comments:

Reviewer's Responses to Questions

**Comments to the Author**

1. If the authors have adequately addressed your comments raised in a previous round of review and you feel that this manuscript is now acceptable for publication, you may indicate that here to bypass the “Comments to the Author” section, enter your conflict of interest statement in the “Confidential to Editor” section, and submit your "Accept" recommendation.

Reviewer #1: (No Response)

2. Is the manuscript technically sound, and do the data support the conclusions?

Reviewer #1: Yes

3. Has the statistical analysis been performed appropriately and rigorously? 

Reviewer #1: Yes

4. Have the authors made all data underlying the findings in their manuscript fully available?

Reviewer #1: Yes

5. Is the manuscript presented in an intelligible fashion and written in standard English?

Reviewer #1: Yes

6. Review Comments to the Author

Reviewer #1: The following issues from my previous review has not yet been addressed:

Lines 46, 169, 177, 267, 276, 285, and 294 - Add a space after the comma.

Line 229 - Spell out what HSE stands for before introducing the acronym.

Lines 356, 375, 391, and 429 - The percentages under LB CI and UB CI should be written as 97.5%, not 97,5%.

Line 357 - In the Notes, the p value should be formatted as:

*p < 0.05; **p < 0.01; ***p < 0.001

instead of:

*p < 0,05; **p < 0,01; ***p < 0,001

Line 376 - In the Notes, the p value should be written as:

p-value < 0.001

instead of:

p-value < 0,001

7. PLOS authors have the option to publish the peer review history of their article (what does this mean? ). If published, this will include your full peer review and any attached files.

**Do you want your identity to be public for this peer review?** For information about this choice, including consent withdrawal, please see our Privacy Policy .

Reviewer #1: No

---

## [Author Response · Author response to Decision Letter 3]

1 Aug 2025

Dear Editor, Dr. Clark,

Dear Reviewers,

We sincerely appreciate your deep engagement with the manuscript and your comments. We hope that we managed to revise the text in line with your suggestions. Below, we respond to them one by one and highlight the changes made in the text.

Reviewer #1:

1. Lines 46, 169, 177, 267, 276, 285, and 294 - Add a space after the comma.

Authors: We added a space under the comma in the according lines.

2. Line 229 - Spell out what HSE stands for before introducing the acronym.

Authors: We spelled out that HSE stands for Higher School of Economics in line 229: The study received approval from the Institutional Review Board at the Higher School of Economics (HSE University). The authors accessed the data on November 28, 2024.

3. Lines 356, 375, 391, and 429 - The percentages under LB CI and UB CI should be written as 97.5%, not 97,5%.

Authors: We have corrected lines 356, 375, 391, and 429, replacing commas with dots.

4. Line 357 - In the Notes, the p value should be formatted as:

*p < 0.05; **p < 0.01; ***p < 0.001

instead of:

*p < 0,05; **p < 0,01; ***p < 0,001

Authors: We made the according correction, replacing commas with dots.

5. Line 376 - In the Notes, the p value should be written as:

p-value < 0.001

instead of:

p-value < 0,001

Authors: We made the correction, replacing commas with dots.

---

## [Editor Report · Decision Letter 3]

5 Aug 2025

Motivation matters: how enrollment motives shape doctoral experiences and career aspirations

PONE-D-25-07628R3

Dear Dr. Smirnov,

We’re pleased to inform you that your manuscript has been judged scientifically suitable for publication and will be formally accepted for publication once it meets all outstanding technical requirements.

Kind regards,

Mary Diane Clark, PhD

Academic Editor

PLOS ONE

Additional Editor Comments (optional):

Thank you for completing the remaining small corrections. Mentoring is an important topic and often lacking for doctoral students.
---

## [Editor Report · Acceptance letter]

PONE-D-25-07628R3

PLOS ONE

Dear Dr. Smirnov,

I'm pleased to inform you that your manuscript has been deemed suitable for publication in PLOS ONE. Congratulations! Your manuscript is now being handed over to our production team.

Kind regards,

on behalf of

Dr. Mary Diane Clark

Academic Editor

PLOS ONE